# Identification of Three Novel Linear B-Cell Epitopes in Non-Structural Protein 3 of Porcine Epidemic Diarrhea Virus Using Monoclonal Antibodies

**DOI:** 10.3390/v16030424

**Published:** 2024-03-09

**Authors:** Mingjun Ye, Huixin Zhu, Zhen Yang, Yanni Gao, Juan Bai, Ping Jiang, Xing Liu, Xianwei Wang

**Affiliations:** 1Key Laboratory of Animal Disease Diagnostics and Immunology, Ministry of Agriculture, MOE International Joint Collaborative Research Laboratory for Animal Health & Food Safety, College of Veterinary Medicine, Nanjing Agricultural University, Nanjing 210095, China; 2021107070@stu.njau.edu.cn (M.Y.); 2018107033@njau.edu.cn (H.Z.); yangzhen@njau.edu.cn (Z.Y.); yngao@njau.edu.cn (Y.G.); baijuan@njau.edu.cn (J.B.); jiangp@njau.edu.cn (P.J.); xingliu@njau.edu.cn (X.L.); 2Jiangsu Co-Innovation Center for Prevention and Control of Important Animal Infectious Diseases and Zoonoses, Yangzhou University, Yangzhou 225009, China

**Keywords:** B-cell epitopes, monoclonal antibody, Nsp3 protein, porcine epidemic diarrhea virus

## Abstract

Porcine epidemic diarrhea virus (PEDV) is a highly pathogenic swine coronavirus that causes diarrhea and high mortality in piglets, resulting in significant economic losses within the global swine industry. Nonstructural protein 3 (Nsp3) is the largest in coronavirus, playing critical roles in viral replication, such as the processing of polyproteins and the formation of replication-transcription complexes (RTCs). In this study, three monoclonal antibodies (mAbs), 7G4, 5A3, and 2D7, targeting PEDV Nsp3 were successfully generated, and three distinct linear B-cell epitopes were identified within these mAbs by using Western blotting analysis with 24 truncations of Nsp3. The epitope against 7G4 was located on amino acids 31-TISQDLLDVE-40, the epitope against 5A3 was found on amino acids 141-LGIVDDPAMG-150, and the epitope against 2D7 was situated on amino acids 282-FYDAAMAIDG-291. Intriguingly, the epitope 31-TISQDLLDVE-40 recognized by the mAb 7G4 appears to be a critical B-cell linear epitope due to its high antigenic index and exposed location on the surface of Nsp3 protein. In addition, bioinformatics analysis unveiled that these three epitopes were highly conserved in most genotypes of PEDV. These findings present the first characterization of three novel linear B-cell epitopes in the Nsp3 protein of PEDV and provide potential tools of mAbs for identifying host proteins that may facilitate viral infection.

## 1. Introduction

Porcine epidemic diarrhea virus (PEDV), a positive-sense single-stranded RNA virus, belongs to the genus *Alphacoronavirus* of the family Coronaviridae [1]. The viral genome is approximately 28 kb and contains seven open reading frames (ORFs). Two polyproteins, PP1a and PP1ab, are encoded by ORF1a and ORF1ab, respectively, which are cleaved into 16 nonstructural proteins (Nsps) mediated by Nsp3 and Nsp5 [2]. Subsequently, the Nsps are assembled into replication-transcription complexes in double-membrane vesicles (DMV), facilitating the replication and transcription of viral genomic RNA (gRNA) [3,4,5]. The remaining ORFs encode spike (S), envelope (E), membrane (M), nucleocapsid (N) proteins, and an accessory protein called ORF3 [6].

PEDV is an acute and highly infectious virus that causes acute watery dysentery, dehydration, electrolyte imbalance, and vomiting, resulting in high mortality in piglets [7,8]. Consequently, it is urgent to develop diagnostic methods to control the disease [8,9,10]. mAbs are widely used in both clinical diagnostics and therapeutics due to their strong specificity and high affinity [11]. Heretofore, mAbs targeting PEDV have predominantly focused on the N and S proteins [12,13,14,15]. The N protein is comparatively conservative across diverse PEDV strains, rendering it a promising candidate as an early diagnostic indicator for PEDV infection [16]. Meanwhile, the S protein has the capacity to stimulate the production of neutralizing antibodies in the host [17,18].

Nsp3 is the largest protein in PEDV, playing a crucial role in various pivotal processes throughout the replication cycle of coronaviruses [19,20], such as the processing of polyproteins [21], the formation of replication-transcription complexes (RTCs) [22], the constitution of double-membrane vesicles (DMV) [23], and the antagonism of host innate immune [24,25]. During SARS-CoV-2 infection, Nsp3, Nsp4, and Nsp6 have been demonstrated to trigger the synthesis of DMVs [26,27]. Subsequently, coronaviruses can exploit DMVs to generate additional RTCs, thereby facilitating the production of an increased quantity of viral RNAs [28]. Additionally, the Nsp3 protein has been predicted to be the second most promising candidate for a vaccine protein after the S protein. This is supported by reports indicating that the Nsp3 protein, functioning as an adhesion protein, plays a crucial role in facilitating viral adherence and entry into host cells [29]. Nevertheless, there is a scarcity of literature regarding the identification of epitopes recognized by mAbs targeting Nsp3 of PEDV. In this study, three mAbs against Nsp3 were successfully screened and three new B-cell antigen epitopes were identified for the first time. These mAbs and identified epitopes may potentially provide valuable tools for the study of the structure and function of Nsp3 protein and serve as a tool to dissect mechanisms of viral replication and identify related host proteins that may aid the virus in hijacking host cells.

## 2. Materials and Methods

### 2.1. Virus, Cells, and Animals

The PEDV strain MSCH (GenBank MT683617.1) was isolated and maintained in our laboratory. Vero cells were cultured in Dulbecco’s Modified Eagle’s Medium (DMEM; Gibco, Grand Island, NY, USA) supplemented with 10% fetal bovine serum (FBS; Bio-channel, Nanjing, China), penicillin (250 U/mL), and streptomycin (250 µg/mL). Mouse myeloma cell lines (SP2/0) were grown in RPMI-1640 (RPMI-1640; Gibco, Grand Island, NY, USA) with 20% FBS. The cells were incubated at 37 °C with 5% CO_2_. BALB/c mice (5–6 weeks old) were procured from the Experimental Animal Center of Yangzhou University.

### 2.2. Gene Cloning, Expression and Purification of Recombinant Nsp3 Protein

The Nsp3 gene was amplified utilizing PEDV cDNA as the template and subsequently cloned into the pET-28a vector through *EcoRⅠ* and *XhoⅠ* restriction sites. Subsequently, the recombinant plasmid, named pET-28a-Nsp3, was transformed into *Escherichia coli* BL21 (DE3) (Vazyme, Nanjing, China) and induced with 0.8 mM isopropyl β-D-1-thiogalactopyranoside (IPTG) overnight at 37 °C. The entire bacterial cells were harvested, suspended in PBS, lysed through sonication, and then centrifuged at 12,000× *g* for 8 min. The proteins were denatured with 8 M urea and purified from the inclusion bodies using His-Sep Ni-NTA Agarose Resin (Amersham Biosciences, Freiburg, Germany). Elution was performed with a gradient of imidazole concentrations, and the final product was assessed through SDS-PAGE and Western blotting analyses. The total protein concentration was determined using a bovine serum albumin protein assay kit (Vazyme, Nanjing, China).

### 2.3. Experiment of Animals

Five BALB/c female mice, aged 5–6 weeks, were purchased and housed at the Animal Experimental Center of Nanjing Agricultural University for a brief observation period. The purified Nsp3 recombinant proteins were mixed with an equal volume of ISA206 adjuvant (SEPPIC, Shanghai, China) and emulsified. Subsequently, each mouse was injected with 60 μg of the protein three times, with a 3-week interval between each immunization. Lastly, blood samples were collected from the tail tips of mice 10 days after the third immunization to measure the serum titers of the immunized mice using indirect ELISA.

### 2.4. Indirect ELISA Assay

An indirect ELISA was established to assess the potency of mice serum and screen positive hybridoma cell lines. Specifically, microtiter plates were coated with purified Nsp3 protein (2 μg/mL) in a carbonated buffer at 4 °C overnight. After blocking with 5% skim milk in PBST at 37 °C for 2 h, hybridoma cultures supernatant and doubly diluted ascites were added into the plates and incubated at 4°C overnight. Simultaneously, the serum of unimmunized and immunized mice (100 μL/well) were employed as negative and positive controls, respectively. Then, a secondary antibody, HRP-conjugated goat anti-mouse IgG (H+L) (Beyotime Biotechnology Co., Ltd., Shanghai, China) (1:1000 dilution), was employed at 37 °C for 1 h. Subsequently, 3,3′-5,5′-tetra-benzidine (TMB) was added for 8 min before the reaction was terminated with 2 M H_2_SO_4_ at room temperature. Finally, the absorbance was measured at 450 nm. A positive result in indirect ELISA was defined as an OD_450_ value exceeding 2.1-fold of the negative controls.

### 2.5. Preparation of mAbs

Immunized mice with the highest antibody titer (1:20,000) determined by indirect ELISA were selected for a booster immunization once the optimal titer was achieved. Two days after the final vaccination, splenic lymphocytes from the immunized mice were collected and fused with SP2/0 cells (at a ratio of 8:1) using PEG4000 (Sigma-Aldrich (Shanghai) Trading Co. Ltd., Shanghai, China). The fused cells were cultured in 2% HAT (Sigma-Aldrich, St. Louis, MO, USA) for seven days and then replaced with 2% HT (Sigma-Aldrich (Shanghai) Trading Co. Ltd.) medium. Positive hybridoma cells, identified by indirect ELISA, were cloned two or three times by limiting dilution. Lastly, hybridoma cells exhibiting higher positive values and good conditions were expanded and preserved in liquid nitrogen tanks. Five-week-old female mice were pre-stimulated with 0.6 mL of liquid paraffin a week in advance. Subsequently, 0.5 mL of hybridoma cells suspended in RPMI-1640 medium were injected into the mice’s enterocoelia to induce the production of ascites. Then, the potency of the hybridoma cell supernatant of every 5 generations and the prepared ascites were evaluated by indirect ELISA. Additionally, the specificity and reactivity of the mAbs were further validated through indirect immunofluorescence assay (IFA) and Western blotting.

### 2.6. IFA

To investigate the specificity and reactivity of the obtained mAbs with PEDV strain MSCH, we conducted an IFA. Vero E6 cells were seeded in 48-well plates and infected with PEDV at a multiplicity of infection (MOI) of 0.1 when the cell density reached 90%. Once the cytopathic effect (CPE) reached 50%, the cell supernatant was substituted with PBS for gently washing three times. Then, cells were fixed with 4% paraformaldehyde for 30 min at room temperature, permeabilized with 0.01% Triton X-100 for 10 min at 37 °C, and subsequently blocked with 5% skimmed milk. After each step, the plates were washed three times with PBS. Next, primary antibodies (anti-Nsp3 and anti-N mAbs) were applied at a dilution of 1:200 in PBS and incubated at 4 °C overnight. The latter mAb served as a positive control to confirm the successful infection of PEDV. After washing with PBS, a secondary antibody, fluorescein isothiocyanate (FITC)-conjugated goat anti-mouse IgG antibody (Proteintech Group, Inc., Rosemont, IL, USA) diluted at 1:200 times, was added and incubated at 37 °C for 45 min. Finally, the treated cells were washed with PBS and observed under an inverted fluorescence microscope.

### 2.7. Western Blotting

PEDV-infected Vero-E6 cells or *E.coli*-expressed Nsp3 proteins were collected at a specific time. Cell supernatants were discarded, and the cells were lysed using RIPA lysis buffer (Beyotime Biotechnology Co., Ltd., Shanghai, China). The lysates or recombinant proteins were electrophoresed on 6% or 10% SDS-PAGE gels, which were dyed with Coomassie brilliant blue or transferred to polyvinylidene difluoride (PVDF) membranes. Following these steps, the membranes were blocked with 3% bovine serum albumin for 2 h at room temperature and then washed with PBST for 30 min. Primary antibodies, anti-His, anti-Nsp3 (hybridoma supernatant), and anti-N mAbs, were utilized to incubate the membranes at 4 °C overnight. Anti-N mAb served as a positive control to confirm PEDV infection. After washing, the membranes were incubated with HRP-labeled goat anti-mouse IgG (diluted at the ratio of 1:5000) for 45 min at 37 °C. Following the final washes, the results were visualized using an ECL Western blotting substrate (Tanon, Nanjing, China).

### 2.8. Isotype Determination of MAbs

The isotype of the anti-Nsp3 mAbs was identified using the mouse monoclonal antibody subtype identification kit (Proteintech Group, Inc., Rosemont, IL, USA), following the manufacturer’s instructions.

### 2.9. Overlapping ELISA

To ascertain whether the three mAbs recognize the identical epitope on Nsp3, we conducted an overlapping ELISA analysis. Firstly, the recombinant Nsp3 protein (2 μg/mL) was coated on the plates. Then, the plates were blocked with 3–5% skimmed milk for 2 h at room temperature. Subsequently, the hybridoma cell culture supernatants were diluted ranging from 1:200 to 1:1,638,400, and then added to the plates to find the optimal concentration of the hybridoma cell culture supernatants for antigen-antibody reaction. The remaining experimental steps were the same as the indirect ELISA. When the OD_450_ values exhibited a linear relationship with the ratio of dilution of cell culture supernatants, we selected the lowest ratio of dilution as the optimal concentration. Secondly, according to the optimal conditions described above, single or two mAbs in different combinations were added in equal amounts to the plates coated with the recombinant Nsp3 protein (2 μg/mL). Lastly, the formula of overlapping coefficient, AI% = [A_1+2_/(A_1_ + A_2_)] × 100%, was used to analyze whether the two antibodies could recognize different antigen binding sites. A_1_ represents the value of OD_450_ when a single mAb reacts independently, while A_2_ represents the value of OD_450_ when another single mAb (whether identical or distinct) reacts independently. When two identical or different mAbs react together, the composite value of OD_450_ was denoted as A_1+2_. The AI% greater than 50% suggests that the two mAbs may recognize different antigen binding sites, whereas the AI% less than 50% indicates that the two mAbs may recognize the same antigen binding site.

### 2.10. Antigen Epitope Analysis

To identify the linear epitopes recognized by the mAbs on the Nsp3 protein, a series of overlapping truncated Nsp3 protein fragments was amplified and cloned into the pET-28a vector. Twenty-four truncated Nsp3 gene fragments were amplified using the primers listed in Table 1. All recombinant truncated proteins were subsequently expressed in *E. coli* BL21 (DE3) and confirmed through Western blotting analysis using the gained mAbs to map the specific mAbs epitope.

### 2.11. Biological Information Analysis

The conservation analysis of epitopes across various PEDV genotypes was performed using BioEdit software (7.0.9.0), utilizing reference sequences sourced from the GenBank database. Furthermore, the spatial features of identified epitopes within the truncated Nsp3 protein were examined by mapping their positions onto a three-dimensional model of the Nsp3 protein through PyMOL software (2.5.0). This methodology enabled the visualization and analysis of Nsp3 protein structures.

## 3. Results

### 3.1. Expression and Purification of the Recombinant Nsp3 Proteins

As illustrated in Figure 1A, the recombinant Nsp3 protein, tagged with His-tag, was effectively expressed in *E. coli* BL21 (DE3), as verified by SDS-PAGE and Coomassie Brilliant Blue staining. Notably, the protein predominantly accumulated in inclusion bodies. Subsequently, purification of the recombinant Nsp3 protein was achieved using different concentrations of imidazole (Figure 1B). The purified protein exhibited a concentration of approximately 0.5 μg/μL (Figure 1C), and the Nsp3 protein displayed strong reactivity with anti-His antibody (Figure 1D).

### 3.2. Preparation of Nsp3 mAbs and Establishment of Genetic Mapping of MAbs

All mice immunized with the Nsp3 protein exhibited a serum antibody titer of at least 1:409,600 (Figure 2A), demonstrating that the targeted protein could elicit a robust immunological response. Mice 1, which produced the highest potency of antibodies against the Nsp3 protein, was selected as the cell fusion source. Three hybridoma cell lines against PEDV Nsp3 protein, namely, 5A3, 7G4, and 2D7, were screened after two to three rounds of subcloning by limiting dilution. Meanwhile, the genetic mapping of mAbs is depicted in Figure 2B. Subsequently, the obtained hybridoma cells were passaged for 25 generations, and the supernatant from every 5 generations was collected and tested by indirect ELISA. As shown in Table 2, the hybridoma cells continued to stably secrete antibodies even after 25 generations. Simultaneously, the antibody titers of ascites from mAbs 7G4, 5A3, and 2D7 reached 1:1,024,000 (Table 2). Furthermore, the subtypes of the aforementioned mAbs were characterized by the mouse monoclonal antibody subtype identification kit, revealing that the light chain of all screened mAbs was kappa, and the heavy chain was IgG1 (Table 2).

### 3.3. Identification of Nsp3 mAbs

To ascertain the characteristics of the acquired mAbs, PEDV-infected and mock-infected Vero-E6 cells were assessed through Western blotting and IFA. Concurrently, the purified Nsp3 protein was employed to validate the reactivity of the mAbs. As depicted in Figure 3A,B, all mAbs exhibited specific reactivity with the recombinant Nsp3 protein and PEDV-infected Vero-E6 cells, distinguishing them from the whole bacteria induced by the pET-28a vector and mock-infected Vero-E6 cells. Additionally, the results from the Western blotting were consistent with IFA, indicating that all mAbs could characteristically recognize PEDV-infected Vero-E6 cells and produce specific green fluorescence (Figure 3C).

### 3.4. Overlapping ELISA for Mapping of the Epitopes

As depicted in Appendix A, the optimal ratios of dilution for the cell culture supernatants of three mAbs, 5A3, 7G4, and 2D7, were 1:800, 1:400, and 1:400, respectively. As shown in Table 3, overlapping ELISA experiments, conducted with the optimal ratio of dilutions of cell culture supernatant determined previously, revealed that the antigenic epitopes recognized by these three mAbs were mutually distinct.

### 3.5. Identification of the Epitopes of PEDV Nsp3 Protein

To facilitate the identification of epitopes targeted by the three mAbs, the Nsp3 protein was initially segmented into three major fragments: F1 (1-151aa), F2 (81-221aa), and F3 (151-301aa). Then, each truncated fraction was expressed and verified using SDS-PAGE and Western blotting. Simultaneously, the Nsp3 protein (1-301aa) and the whole bacteria served as positive and negative controls, respectively. As illustrated in Figure 4C, mAb 5A3 recognized both F1 and F2 fragments but not the F3 segment, pinpointing the antigenic epitope of 5A3 to amino acids 81-151. In Figure 4B,D, mAb 7G4 exclusively interacted with the F1 segment, while mAb 2D7 recognized the F3 segment. This indicated that the epitope regions of 7G4 and 2D7 were located at amino acids 1-81 and 221-301, respectively. To further narrow down the epitopes, 21 truncated Nsp3 proteins were expressed, and SDS-PAGE and Western blotting were performed to verify their expression (Figure 4(Aa,Ab)). Subsequently, in combination with the approximate location range of the three mAbs, strategies for identifying antigen epitopes were outlined in Figure 4B–D. MAb 7G4 recognized F1, F8, F9, F10, and F11 fragments while not interacting with F2, F3, F4, F5, F6, and F7 segments. MAb 5A3 reacted with F1, F2, F12, F13, F14, F15, F16, and F17 fragments except for the F3 segments. Similarly, mAb 2D7 interacted with F3, and F24 fragments but not F18, F19, F20, F21, F22, and F23 segments. These findings demonstrated that the antigenic epitope regions of mAbs 7G4, 5A3, and 2D7 were potentially situated at amino acids 31-41, 141-151, and 281-291, respectively. In conclusion, the epitope targeting 7G4 and 5A3 existed at ^31^TISQDLLDVE^40^ and ^141^LGIVDDPAMG^150^, respectively, while mAb 2D7 recognized the epitope ^282^FYDAAMAIDG^291^.

### 3.6. Homology Analysis of Epitope of Different PEDV Genotypic Strains

To evaluate the conservation of the identified epitopes recognized by the screened mAbs across different genotypic PEDV strains, the amino acid sequences of all identified epitopes were aligned using BioEdit software. The results revealed that the three identified epitopes are highly conservative among diverse PEDV isolates (Figure 5A). Epitopes ^31^TISQDLLDVE^40^ and ^282^FYDAAMAIDG^291^ were found to be conserved in most PEDV isolates, with “I32” in epitope ^31^TISQDLLDVE^40^ being occasionally replaced by “L32” or “34Q” being substituted by “H34”. Similarly, “283F” in epitope ^282^FYDAAMAIDG^291^ was replaced by “283Y”. However, in contrast to the three PEDV MSCH, YZ, and SH strains, the amino acid “G150” in the epitope ^141^LGIVDDPAMG^150^ was substituted by “E150” in other PEDV strains.

### 3.7. Prediction of the Spatial Structures and Analysis of Epitopes

The prediction and analysis of the spatial structure of proteins can greatly aid in understanding the antigenicity of relevant epitopes. Therefore, it is essential to conduct a thorough analysis of spatial structures in protein. The amino acid sequence of Nsp3 was submitted to the Robetta website to predict its spatial structure, and the identified epitopes were subsequently located in Figure 5B. As depicted in Figure 5C, the epitope ^31^TISQDLLDVE^40^ (highlighted in orange) against 7G4 maintained a high antigenic helix. Conversely, the analysis of hydrophilicity indicates that the epitope ^141^LGIVDDPAMG^150^ (marked in green) against 5A3 and the epitope ^282^FYDAAMAIDG^291^ (highlighted in yellow) against 2D7 may be situated within the internal regions of the spatial structure. These results may be crucial for elucidating the function and structure of Nsp3 protein.

## 4. Discussion

Since the onset of the PEDV epidemic, efforts have been primarily centered on developing vaccines and methods of detection targeting the N or S proteins to mitigate the spread of the virus. Although some progress has been made in inactivated and attenuated PEDV vaccines [30,31], swine herds vaccinated with these vaccines still encounter heightened mortality rates among newborn piglets [32,33]. This could be linked to the relatively short duration of immunization with inactivated vaccines and the resurgence of virulence from live attenuated vaccines. Hence, it is crucial to research the next generation of vaccines, such as peptide-based vaccines, and develop potential targets for prophylactic or therapeutic vaccines to prevent outbreaks of novel PEDV strains. To date, the Nsp3 protein of PEDV has not been utilized as a coronavirus vaccine target. Nevertheless, the non-structural protein of the hepatitis C virus has been demonstrated to induce strong and effective HCV-specific immunity [34], and the gene products of non-structural protein targeting HIV-1 have been shown as valuable targets for designing prophylactic or therapeutic vaccines [35]. While structural proteins have traditionally been the primary focus of viruses, some studies suggested that nonstructural protein 3 of COVID-19 could be a promising vaccine candidate, which could be capable of inducing cellular or humoral immunity to suppress viral invasion or replication [29]. Further investigation is needed to determine whether the Nsp3 protein of PEDV has similar functions.

Innate immunity, serving as the primary line of host defense against viral infection, triggers a series of signaling cascades to induce the expression of antiviral genes by recognizing pathogen-associated molecular patterns (PAMPs) through host pattern recognition receptors (PRRs) [36,37]. The extensive replication of the virus within host cells necessitates the suppression or evasion of the host’s interferon (IFN) response. This process is typically mediated by various viral proteins operating through diverse mechanisms [38]. The Nsp3 protein of SARS-CoV-2 contains a macrodomain (Mac1) that can hydrolyze ADP-ribose modifications, leading to the suppression of the antiviral innate immune response [39]. Additionally, the papain-like protease (PLP) domain of Nsp3 of SARS-CoV plays a role in downregulating the activation of IRF3 in the innate immune response by interacting with the STING-TRAF3-TBK1 complex, providing a mechanism for SARS-CoV to evade the host’s innate immune response [40,41]. The PLP domain of MERS-CoV acts as a viral deubiquitinating enzyme (DUB) and an IFN antagonist, inhibiting the production of IFN-β by interfering with the phosphorylation and nuclear translocation of IRF3 [42]. Similarly, the PLP domain of HCoV-NL63 suppresses the activation of IRF3 and the transcription of IFN-β by targeting and eliminating STING [43]. The PLP2 domain of Nsp3 of MHV-A59 exhibits dual functions by deubiquitinating TBK1, leading to the inactivation of its kinase activity for phosphorylating IRF3, and by delaying the dissociation of IRF3 from TBK1, which could effectively attenuate IFN induction [44]. Additionally, the Nsp3 (amino acids 590-1215) of TGEV has been shown to inhibit NF-κB signaling by suppressing the ubiquitination of IκBα and restricting the phosphorylation and nuclear translocation of p65 [45]. Furthermore, the PLP domain of IBV has the ability to remove ubiquitin modifications from proteins involved in antiviral innate immune pathways, ultimately leading to the blockage or delay of the host innate immune response in IBV-infected cells [46]. Likewise, the PLP2 of PEDV functions as a viral IFN antagonist and a DUB. By cleaving the ubiquitin chains of RIG-I and STING, it effectively inhibits the activation of type I IFN signaling transduction [25]. To better comprehend the antagonistic effect of the Nsp3 protein of coronavirus on innate immune responses and to develop antiviral drug development, this study selected the Nsp3 protein of PEDV as the target for preparing mAbs, providing foundational support for further exploring the interaction between Nsp3 protein of PEDV and the antiviral innate immune response.

To a certain extent, research based on antigenic epitopes has significantly contributed to the development of epitope-based vaccines, aiming to discover potential therapeutic targets. By pinpointing specific antigenic epitopes, researchers can design vaccines targeting these key areas, potentially resulting in more effective and targeted immune responses against viral infections. Currently, the majority of studies on antigenic epitopes have concentrated on the S and N proteins of PEDV. In contrast, research on the Nsp3 protein has primarily focused on the development of targeted drugs rather than the identification of antigenic epitopes. Additionally, linear B-cell epitopes on the Nsp3 protein of PEDV have been sparsely identified. In this study, we report three novel linear B-cell epitopes of Nsp3 for the first time, which may catalyze a deeper investigation into the function of the Nsp3 protein of PEDV and assist in identifying host proteins that can facilitate the virus in hijacking host cells.

To further investigate the specificity of the obtained mAbs, we employed Western blotting and IFA to characterize the reactivity of the screened mAbs with PEDV and recombinant Nsp3 protein. MAbs, 7G4, 5A3, and 2D7, exhibited specific reactivity with the Nsp3 protein expressed in the prokaryotic system and PEDV-infected Vero cells. However, the presence of multiple bands in the Western blotting from the lane of PEDV-infected cells, particularly the band around 180 kDa, aligned with the expected molecular weight of the Nsp3 protein. The presence of additional bands may indicate various cleavage forms of the Nsp3 protein during PEDV replication. This finding not only shed light on the complexity of Nsp3 protein during viral replication but also underscored the dynamic nature of viral protein expression and modification within infected cells.

To pinpoint the epitopes of the screened mAbs, we utilized the method of classical truncated protein expression to recognize the epitopes by mAbs. Three entirely novel antigenic epitopes were characterized, including ^31^TISQDLLDVE^40^, ^141^LGIVDDPAMG^150^, and ^282^FYDAAMAIDG^291^. Alignment of the amino acid sequences of different PEDV strains revealed that the three identified antigenic epitopes were relatively conservative. To further analyze the spatial distribution of the characterized epitopes, PyMol software was used to assess whether the epitopes were present on the surface of the protein. It was revealed that ^31^TISQDLLDVE^40^ was completely exposed on the surface of the Nsp3 protein, maximizing possibilities for antigen-antibody interaction.

In summary, we successfully generated three mAbs against the Nsp3 protein of PEDV and identified three novel linear B-cell epitopes for the first time. This discovery offers opportunities to delve into the structure and function of the Nsp3 protein of PEDV and identify host proteins that play a role in aiding the virus in hijacking host cells.

## Figures and Tables

**Figure 1 viruses-16-00424-f001:**
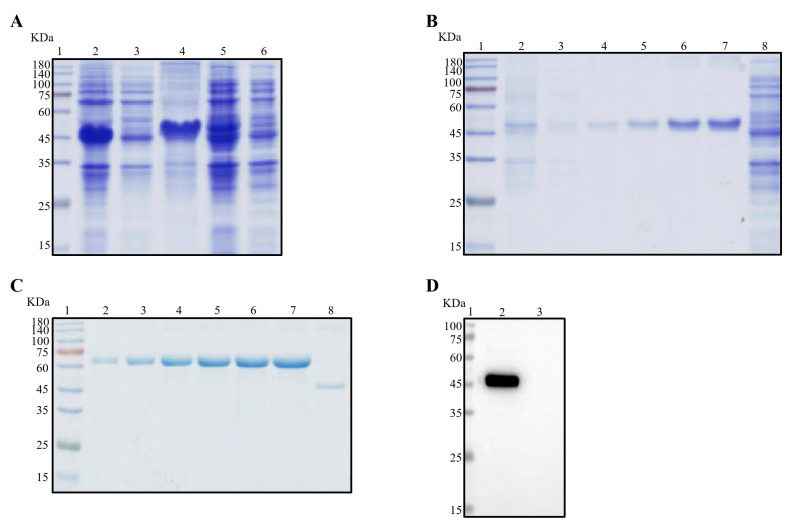
Expression and purification of Nsp3 proteins of PEDV with His-tag. Bacterial lysates from *E. coli* BL21 with recombinant plasmids pET-28a-Nsp3 were subjected to SDS-PAGE (**A**–**C**) and Western blotting (**D**) analysis with anti-His mAb. The protein ladder was in Lane 1. (**A**) Lane 2, total bacterial cell lysates after induction by IPTG; Lane 3, the supernatant of BL21 with pET-28a-Nsp3 induced by IPTG after sonication; Lane 4, the precipitation of BL21 with pET-28a-Nsp3 induced by IPTG; Lane 5, the whole bacterial cell lysates without induction by IPTG; Lane 6, negative control with pET-28a vector. (**B**) Lanes 2 to 7 contained eluents with imidazole ranging from 20 to 250 mM; Lane 8, the entire bacteria BL21 with pET-28a vector stimulated by IPTG. (**C**) The concentrations of bovine serum albumin (BSA) in Lanes 2 to 7 ranged from 0.5 μg/μL to 3.0 μg/μL; Lane 8, the purified Nsp3 protein. (**D**) Lane 2, the purified Nsp3 protein; Lane 3, pET-28a vector serving as the negative control. Anti-His mAb was employed as the primary antibody, and horseradish-peroxidase-labeled goat anti-mouse IgG antibody was utilized as the secondary antibody in Western blotting.

**Figure 2 viruses-16-00424-f002:**
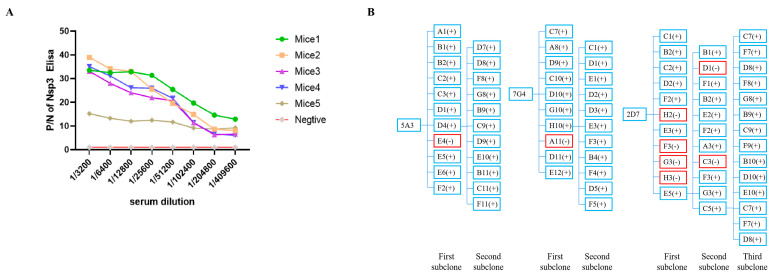
Determination of serum antibody potency against Nsp3 protein in mice after three immunizations and establishment of genetic mapping of mAbs. (**A**) Blood samples were collected from the tail tips of five mice, and the serums were isolated. Serial two-fold diluted serum samples were then analyzed using indirect ELISA, with the purified Nsp3 protein as the coating antigen. Serum samples from unimmunized mice served as the negative control. The absorbance value at OD_450_ nm of serum from immunized mice was considered as the positive value (P), while the value from unimmunized mice was designated as the negative value (N). (**B**) The symbol “+” in blue indicated a positive monoclonal clone, while “-” in red represented a negative monoclonal clone.

**Figure 3 viruses-16-00424-f003:**
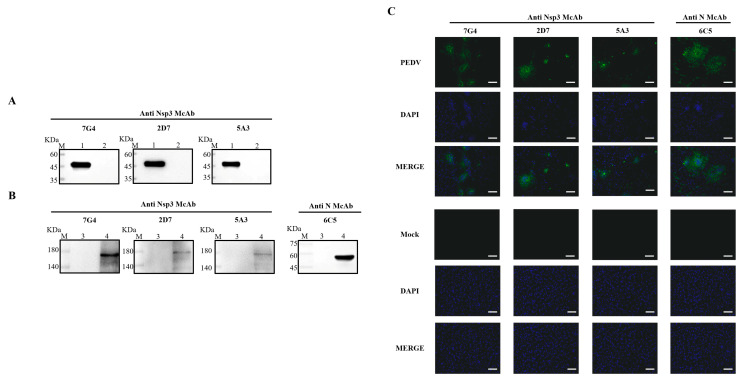
Identification and characterization of the mAbs. The lysates from Vero-E6 cells infected with PEDV or Nsp3 protein produced in *E. coli* BL21 were collected for SDS-PAGE analysis, while the lysates from PEDV-uninfected Vero-E6 cells and *E. coli* BL21 were utilized as corresponding controls, respectively. (**A**,**B**) M represented a protein ladder; Lane 1, the purified Nsp3 protein; Lane 2, the negative control with the pET-28a vector; Lanes 3 and 5, the lysates from PEDV-uninfected Vero-E6 cells; Lanes 4 and 6, the lysates from PEDV-infected Vero-E6 cells. MAbs 7G4, 2D7, and 5A3 were used as primary antibodies, and the goat anti-mouse IgG antibody labeled with horseradish peroxidase was utilized as the secondary antibody. (**C**) The mAbs were identified by IFA in Vero-E6 cells infected with PEDV (scale bar: 100 μm). Vero-E6 cells were infected with PEDV or mock-infected at 0.1 MOI for 12 h. After fixing, permeabilizing, and blocking, the Vero-E6 cells were incubated with the mAbs at 4 °C overnight. The secondary antibody was goat anti-mouse IgG antibody (H+L) conjugated to fluorescein isothiocyanate (FITC). Fluorescence signals were observed in Vero-E6 cells infected with PEDV using mAbs 7G4, 2D7, 5A3, and N. MAb N was utilized as a positive control to ensure the successful infection of PEDV.

**Figure 4 viruses-16-00424-f004:**
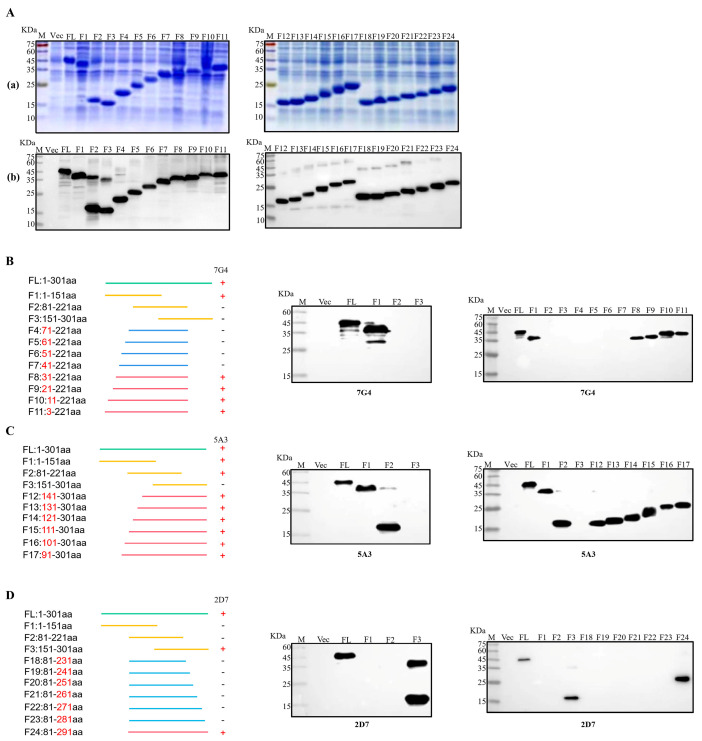
Identification of B-cell epitopes recognized by mAbs. (**Aa**) Twenty-four fragments overlapping Nsp3 were expressed in *E. coli* BL21 and analyzed by SDS-PAGE. (**Ab**) Twenty-four fragments overlapping Nsp3 were expressed in *E. coli* BL21 and analyzed by Western blotting with mAb anti-His to confirm the expression of all truncated proteins. (**B–D**) Multiple fragments overlapping Nsp3 were expressed in *E. coli* BL21 and identified by Western blotting using mAbs 7G4, 5A3, and 2D7.

**Figure 5 viruses-16-00424-f005:**
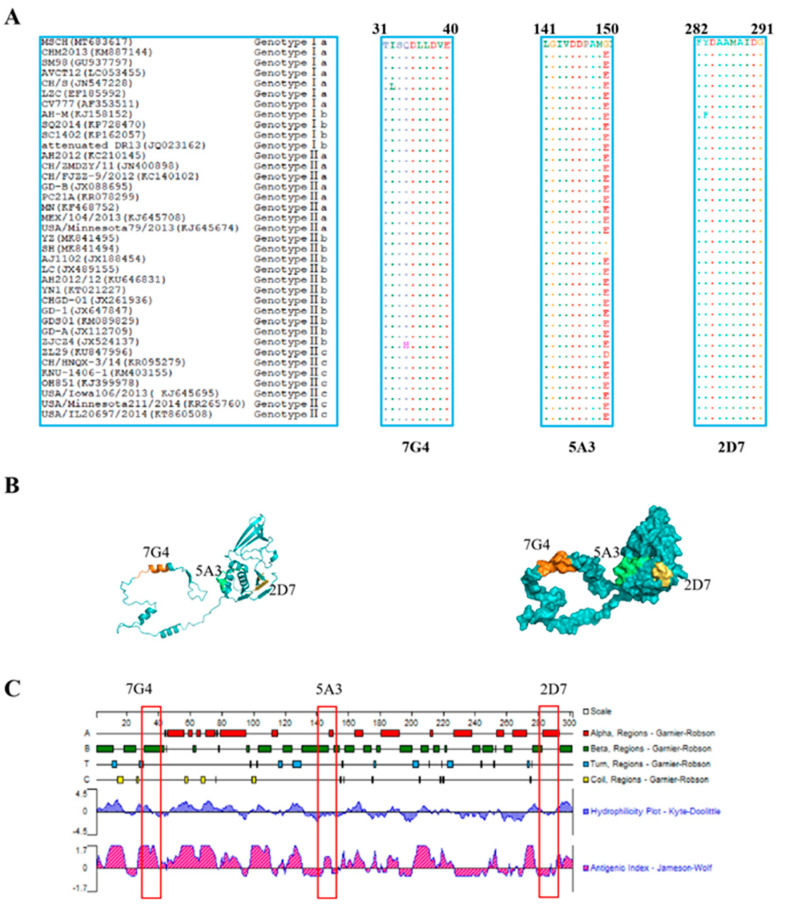
Character analysis of epitopes identified by mAbs on the Nsp3 proteins. (**A**) Sequences comparison of the identified epitopes, ^31^TISQDLLDVE^40^, ^141^LGIVDDPAMG^150^, and ^282^FYDAAMAIDG^291^, among different genotypes of PEDV strains, were downloaded from GenBank. The amino acid sequences of the identified epitopes from different PEDV strains were aligned using the BioEdit software, and the identical amino acid residues were represented by “ˑ”. (**B**) The prediction of the structure of the Nsp3 protein was performed through the Robbeta website, and the epitopes of the Nsp3 protein were located in a simulated 3D model using the PyMol software. The epitopes against 7G4 (marked in orange), 5A3 (highlighted in green), and 2D7 (marked in yellow) were highlighted with different colors. (**C**) The antigenicity and surface probability of Nsp3 protein were analyzed by Protean software (7.1.0). The red box indicates the locations of the antigenic epitopes identified for the three monoclonal antibodies.

**Table 1 viruses-16-00424-t001:** Primers of genes for PCR amplification.

Primers	Sequence (5′-3′)	Production Size (bp)
Nsp3-F	CCGAATTCATGGATGTGCCTAAGTACTACA	903
Nsp3-R	CGCTCGAGTGTGTCATACTTTATCTGATGAC	
Nsp3-F	CCGAATTCATGGATGTGCCTAAGTACTACA	453
F1-R	CCCTCGAGAAGCCCCATTGCAGG	
F2-F	CCGAATTCACAAATGTAGAGTCTGAAGTT	420
F2-R	CCCTCGAGTTCCAAAGTTGGCGTCAT	
F3-F	CCGAATTCCTTTTTAGTGCTGGTAGAGTT	450
Nsp3-R	CGCTCGAGTGTGTCATACTTTATCTGATGAC	
F4-F	CCGAATTCGAGGATGACGGTCTTAAT	450
Nsp3-R	CGCTCGAGTGTGTCATACTTTATCTGATGAC	
F5-F	CCGAATTCAAGGTGGCAGACGTG	480
Nsp3-R	CGCTCGAGTGTGTCATACTTTATCTGATGAC	
F6-F	CCGAATTCGGTGATGAAGTAGACTCC	510
Nsp3-R	CGCTCGAGTGTGTCATACTTTATCTGATGAC	
F7-F	CCGAATTCGTTGTTACTGATGCGC	540
Nsp3-R	CGCTCGAGTGTGTCATACTTTATCTGATGAC	
F8-F	CCGAATTCACGATCTCACAGGATCTG	570
Nsp3-R	CGCTCGAGTGTGTCATACTTTATCTGATGAC	
F9-F	CGGAATTCATGGTTTCTCAGTGGC	600
Nsp3-R	CGCTCGAGTGTGTCATACTTTATCTGATGAC	
F10-F	CCGAATTCGAAGGTGGCACCGAT	630
Nsp3-R	CGCTCGAGTGTGTCATACTTTATCTGATGAC	
F11-F	CCGAATTCCCTAAGTACTACATCTATGATGAG	660
Nsp3-R	CGCTCGAGTGTGTCATACTTTATCTGATGAC	
F12-F	CCGAATTCCTTGGCATCGTTGATGAC	480
Nsp3-R	CGCTCGAGTGTGTCATACTTTATCTGATGAC	
F13-F	CCGAATTCGTTACTTCTACCTTGGTG	510
Nsp3-R	CGCTCGAGTGTGTCATACTTTATCTGATGAC	
F14-F	CCGAATTCGTTTTAAGACAATCTCATAACAAC	540
Nsp3-R	CGCTCGAGTGTGTCATACTTTATCTGATGAC	
F15-F	CCGAATTCTTTGACTTTGCAAGCTAT	570
Nsp3-R	CGCTCGAGTGTGTCATACTTTATCTGATGAC	
F16-F	CCGAATTCCCTTCCACAGTTACTAAGGAT	600
Nsp3-R	CGCTCGAGTGTGTCATACTTTATCTGATGAC	
F17-F	CCGAATTCGCCGCAACCTTGTCC	630
Nsp3-R	CGCTCGAGTGTGTCATACTTTATCTGATGAC	
F2-F	CCGAATTCACAAATGTAGAGTCTGAAGTT	450
F18-R	CCCTCGAGACACTGAGCACAAGC	
F2-F	CCGAATTCACAAATGTAGAGTCTGAAGTT	480
F19-R	CCCTCGAGACTTTTAAAAGTGTGCATCAA	
F2-F	CCGAATTCACAAATGTAGAGTCTGAAGTT	510
F20-R	CCCTCGAGATCTCGACAAAAGATGCC	
F2-F	CCGAATTCACAAATGTAGAGTCTGAAGTT	540
F21-R	CCCTCGAGAACCAAAGAATCCAAGGA	
F2-F	CCGAATTCACAAATGTAGAGTCTGAAGTT	570
F22-R	CCCTCGAGTATAAAAGCAGCCGCACA	
F2-F	CCGAATTCACAAATGTAGAGTCTGAAGTT	600
F23-R	CCCTCGAGGTTAGTGACATAATGACCACT	
F2-F	CCGAATTCACAAATGTAGAGTCTGAAGTT	630
F24-R	CCCTCGAGACCATCAATAGCCATAGC	

**Table 2 viruses-16-00424-t002:** Isotype identification and antibody titers of mAbs.

Monoclonal Antibody	Antibody of Hybridoma Cells Cultural Supernatant Potency					Ascites Potency	Heavy Chain	Light Chain
	F5	F10	F15	F20	F25			
5A3	1:204,800	1:204,800	1:204,800	1:204,800	1:204,800	1:102,4000	IgG1	Kappa
7G4	1:12,800	1:12,800	1:12,800	1:25,600	1:25,600	1:102,4000	IgG1	Kappa
2D7	1:3200	1:3200	1:3200	1:3200	1:3200	1:102,4000	IgG1	Kappa

**Table 3 viruses-16-00424-t003:** Analysis of overlapping coefficient of mAbs.

Monoclonal Antibodies	OD_450_ and the Overlapping Coefficients		
	5A3	7G4	2D7
5A3	2.145 (50%)	2.632 (54%)	2.326 (59%)
7G4	2.830 (58%)	2.726 (50%)	2.677 (59%)
2D7	2.415 (61%)	2.630 (58%)	1.784 (50%)

## Data Availability

All data created in this study has already been shown in the manuscript. No more new data is available.

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
