# Peer review of "Identification of Three Novel Linear B-Cell Epitopes in Non-Structural Protein 3 of Porcine Epidemic Diarrhea Virus Using Monoclonal Antibodies"

_viruses, 2024, doi:10.3390/v16030424_

Round 1

Reviewer 1 Report (Previous Reviewer 2)

Comments and Suggestions for Authors

This is a well-written paper containing interesting results which merit publication. For the benefit of the reader, however, a number of points need clarifying and certain statements require further justification. There are given below.

Line 16: antibodies(mAbs) needs space in front of (.

Line 36: what is gRNA?

Line 44: protains. [11-14]. Delete priod.

In the manuscript, authors use both long and short bar. For example, line 113-114. Please use short bar.

Line 116: delete , Shanghai, China.

 Line 129: Infection (MOI) infection

Line 138: antibody(Proteintech needs space.

Line 138: times) → times

Line 143-144: Lysis Buffer  → lysis buffer

Line 156-157: Mouse Monoclonal Anti- body Subtype Identification Kit, please use small letter.

Line 171: protein(2 → protein (2

In Table 1:  the letter of F24-R primer is small.

Line 224: Bovine Serum Albumin → bovine serum albumin

Figrue legends: letter size is different in the figurev legends. Please changes.

Figure 4: Do authors need this figre 4?  As a Supplementary data is better.

Line 360: the letter size of 6 is small.

Author Response

Reviewer 2 Report (New Reviewer)

Comments and Suggestions for Authors

Porcine Epidemic Diarrhea Virus (PEDV) is a highly contagious alphacoronavirus that primarily affects swine, leading to significant mortality, especially in suckling piglets, and positioning it as a major global  concern due to its economic impact on the pork industry. Research into the viral biology of PEDV is ongoing, focusing on understanding its pathogenesis, transmission dynamics, and immune evasion mechanisms, with the aim of developing effective control measures including vaccines or other preventative interventions. The authors of this submission have developed three new monoclonal antibodies targeting PEDV Nsp3, further presenting convincing validation and mapping of the epitopes targeted. These antibodies could be useful research tools employed to investigate a less-studied viral protein with potential importance in virulence and immune evasion.

It is interesting to note that the epitopes identified are highly conserved across isolate genomes examined. This is desirable for broadly useful research tools, but would suggest that these may not be primary B cell epitopes targeted by the host animal immune response, due to the lack of observed diversifying mutation under selective pressure for immune invasion -- a quality observed to a great degree in the better-studied spike protein antigen.

I suggest adding some further discussion of existing research into Nsp3 function (for example in mediating interferon response, in immune evasion, and into its enzymatic activities), to strengthen the interest and impact of this work in the context of the investigations it may aid.

I suggest making Figure 3c images brighter. 

Comments on the Quality of English Language

Minor editing for word choice is recommended in a few places.

Round 2

Reviewer 1 Report (Previous Reviewer 2)

Comments and Suggestions for Authors

The figure needs to be revised. For example, Fig. 5C is not visible and is cut off.

This manuscript is a resubmission of an earlier submission. The following is a list of the peer review reports and author responses from that submission.

Round 1

Reviewer 1 Report

Comments and Suggestions for Authors

The manuscript described the generation and characterization of three monoclonal antibodies to the NSP3 of PEDV. The authors concluded three novel linear B-cell epitopes were identified in NSP3. There are numerous spelling and grammar errors throughout the manuscript. Some figures are of poor quality for publication. Additional experiments are needed to validate the conclusions of the paper. See specific comments below.

1. Fig.3B and 3C are of poor quality. Fig.3C needs to include a non-infected cells as controls and DAPI staining to see the cellular localization of the specific staining. Fig3B needs a better-quality picture.

2. It is recommended that the authors synthesize over-lapping peptides and conduct peptide ELISA assays to fully validate three linear epitopes.

3. The authors need to better describe the rationale and significance of the study and the future applications of the findings.

Comments on the Quality of English Language

There are numerous spelling and grammar errors throughout the manuscript.

Reviewer 2 Report

Comments and Suggestions for Authors

This is a well-written paper containing interesting results which merit publication. For the benefit of the reader, however, a number of points need clarifying and certain statements require further justification. There are given below.

1.         The content of Figure 4 and the description of the result (L252-275) do not match and need to be corrected.

2.         The description of the results (L306-313) does not match the content of Fig. 5, and needs to be corrected.

3.         L395: An addition is necessary because there is no permission number for animal experiments.

4.         The references are doubly numbered.

5.         Abbreviations are specified in the text but are spelled out in some places or go, which needs to be corrected. For example, immunofluorescence assay (IFA).

6.         L27: Keywords are in alphabetical order.

7.         L34: Nsp3 (papain-like protease (PLP)) is complicated, is PLP necessary?

8.         L34: Nsp5 34 (3C-like protease (3C)) is similar.

9.         L73: CO2?

10.      L91: 206 adjuvant, is this correct?

11.      L206: Fig. 2B -> Fig. 2B

Reviewer 3 Report

Comments and Suggestions for Authors

I'm afraid I don't see the purpose of this research. Authors said it will offer a beneficial tool for disease diagnosis and vaccine development in the future, but there is no problem to diagnose PEDV worldwide and vaccine development requires more sophisticated approach rather than what this research proposed.

It seems just that authors have been doing lab work that they have been doing so far, got any results, and then fit the purpose to the results. No clear aim of this research could be found.

Also, in the material and method section, it is written in terms commonly used for convenience in lab without consistent rules to describe materials including company, city, nation.

In the discussion section, the content is elaborated in a way that seems somewhat unrelated to the research results, raising doubts about whether the authors have properly contemplated the study. 

Comments on the Quality of English Language

English is fairly good, but there are small grammatical errors throughout, which need to be corrected.